# Connecting degree and polarity: An artificial language learning study

**Lisa Bylinina**
University of Groningen
e.g.bylinina@rug.nl

**Alexey Tikhonov**
Inworld.AI, Germany
altsoph@gmail.com

**Ekaterina Garmash**
Spotify, United Kingdom[*]
katya.garmash@gmail.com

## Abstract

We investigate a new linguistic generalisation in pre-trained language models (taking BERT Devlin et al. 2019 as a case study). We focus on degree modifiers (expressions like *slightly*, *very*, *rather*, *extremely*) and test the hypothesis that the degree expressed by a modifier (low, medium or high degree) is related to the modifier's sensitivity to sentence polarity (whether it shows preference for affirmative or negative sentences or neither). To probe this connection, we apply the Artificial Language Learning experimental paradigm from psycholinguistics to a neural language model. Our experimental results suggest that BERT generalizes in line with existing linguistic observations that relate degree semantics to polarity sensitivity, including the main one: low degree semantics is associated with preference towards positive polarity.

## 1 Introduction

Linguistic expressions can be characterized along a variety of properties: what they mean, what parts they consist of, how they combine with other expressions and so on. Some of these properties are systematically related to each other. When these relations appear systematically in language after language, they can be grounds for implicational linguistic universals, for example, Greenberg's **Universal 37**: *A language never has more gender categories in nonsingular numbers than in the singular.* (Greenberg, 1963). Here, two properties of linguistic expressions are related: the grammatical number of an expression and how many gender distinctions are available for this expression. More complex generalizations may concern correlation between continuous properties $A$ and $B$.

In order to arrive at linguistic universals connecting $A$ and $B$, the relation between these properties has to be established at the level of individual languages, which is not trivial for properties with complex internal structure.

In this paper, we study one such connection: the problem of polarity sensitivity of degree modifiers (Israel, 1996, 2011; Solt, 2018; Solt and Wilson, 2021). Degree modifiers are words like *slightly*, *very*, and *extremely*. Property $A$, in this case, is the **degree** that these words convey, defined on a interval from very low to very high. For example, the degree of *slightly* is lower than the one of *very*. Property $B$ here encodes distributional preferences of degree modifiers with respect to **polarity** of a sentence where they appear – roughly, whether they appear exclusively in negative or affirmative sentences, or show no polarity preference. Polarity preferences can also be represented as a continuous property from very low (negative polarity preference) to very high (positive polarity preference), with polarity-neutral in the middle.

Interactions between linguistic properties have been subject to experimental studies in psycholinguistics and cognitive science. One prominent method is *Artificial Language Learning* (Friederici et al., 2002; Motamedi et al., 2019; Kanwal et al., 2017; Culbertson et al., 2012; Ettlinger et al., 2014; Finley and Badecker, 2009). It has the following main ingredients:

1. **fragment of an artificial language** in the form of expressions that do not belong to the language that participants are speakers of;

2. **training phase**, where some information about the language fragment is given to the participants;

3. **testing phase**, where it is checked what other knowledge, beside the provided, was inferred during training.

Originally designed for studies with human participants, the Artificial Language Learning framework has also been applied to neural network-based learning models (Piantadosi et al., 2012; Carcassi et al., 2019; van de Pol et al., 2021). Replacing

---

[*] Work was performed previously while at Huawei.

human participants with artificial learning agents allows to examine the learning process in more detail and to make a variety of learnability statements. One important property of these experiments is that the learning agents typically come in a blank state with no prior knowledge or biases. This limits the set of linguistic questions that can be targeted by this type of experiment.

The way we use the Artificial Language Learning paradigm can be seen as middle ground between experiments with human participants and with artificial learners described above. Our approach also involves an artificial language fragment and a training procedure to introduce knowledge about some property $A$, but it uses a *pre-trained* language model (LM) (Peters et al., 2018; Devlin et al., 2019; Brown et al., 2020) as the learning agent. More technically, we extend a pre-trained LM with a set of new tokens with randomly initialized embeddings and perform fine-tuning on a carefully constructed synthetic dataset. The dataset is constructed in a way to indirectly introduce different values along property $A$ for different new tokens. Upon fine-tuning, we measure how the training affected property $B$ and how variation along $B$ depends on the values of property $A$ introduced during training.

Our study will focus on English, as represented in a pre-trained LM BERT (Devlin et al., 2019). We see this as a proof of concept work that can be extended further along the cross-linguistic dimension and in application to other models. Using this set-up, we address the question of whether LMs encode a connection between degree semantics and polarity, therefore making a generalization across two different linguistic properties.

Our approach belongs to the general area of studies using counterfactual linguistic data in NLP (Kaushik et al. 2020, 2021 a.o.); a part of that subfield that uses novel lexicon is the closest to the present paper (Thrush et al. 2020; Bylinina and Tikhonov 2022a a.o.). Our work contributes to the general agenda of establishing closer connections between learning in humans and artificial neural models (Futrell et al. 2019; Wilcox et al. 2020 a.o.). The main step forward that we make with this paper is the extension of these methods to linguistic properties that have complex internal structure, rather than clear morphological or syntactic exponence.

To sum up, we make the following contributions:

- We propose an experimental methodology to explore generalization between complex linguistic properties;

- We use this methodology to explore the relation between two linguistic phenomena, degree and polarity sensitivity, as represented in one pre-trained LM (BERT).

- We argue that, according to the experimental results, the LM in question indeed makes a connection between the degree encoded by a degree modifier and its polarity sensitivity.

The paper is structured as follows: Section 2 gives linguistic background about degrees and polarity. Section 3 describes the general method. In Section 4, we define a synthetic dataset and the measures we use to estimate degree and polarity. Section 5 presents the experiment. Section 6 discusses our results, the limitations of our set-up and suggestions for future work.

## 2   Background: Degree and polarity

In this section we provide background on the studied linguistic properties: we describe *degree* as a property of degree modifiers, and *polarity sensitivity* as a property of linguistic items (words) that tend to appear in certain types of contexts. We outline the relation between these two properties, as discussed in theoretical linguistic literature. We will apply our proposed method to experimentally verify the hypothesised relation.

**Degree**

So-called gradable adjectives describe properties that can hold to a different degree. A classic example of a gradable adjective is *tall*. A classic example of a non-gradable one is *prime*. The former, as opposed to the latter, can be part of comparative and superlative constructions, and they can combine with **degree modifiers**: words like *slightly*, *very*, and *extremely*. Examples (1)-(2) illustrate this difference. We use * to denote all types of linguistic deviancy, including ungrammaticality as well as semantic / pragmatic oddity:

(1)    *7 is more prime than 3.
       *13 is the most prime number in this set.
       *1 is **somewhat / very / extremely** prime.

(2)    Mary is taller than John.
       Mary is the tallest person in this room.
       Mary is **somewhat / very / extremely** tall.

For a statement with a simple base form of a gradable adjective – like *Mary is tall* – to be true, the property in question has to hold of the subject to some degree $\delta$ that is determined by linguistic and extra-linguistic contextual factors (Fara, 2000; Kennedy and McNally, 2005; Kennedy, 2007). When a gradable adjective appears in combination with a degree modifier, the degree $\delta$ that makes the statement true changes to a value that depends on the modifier. For Mary to count as 'somewhat tall', her height needs to be much lower than for 'extremely tall', for instance. The requirements on $\delta$ that degree modifiers encode can be used to order these modifiers along a scale of degrees, for example, *somewhat < extremely*.

In our experiments, we will map degree modifiers to a dense interval from 0 to 1 according to their degree semantics (from very low to very high).

**Polarity sensitivity**

For certain expressions, their acceptability and/or interpretation in a context is conditioned on the polarity of this context. Expressions with distributional preference[1] for negative contexts are called negative polarity items (NPIs). Expressions with preference towards positive contexts are called positive polarity items (PPIs). For example, *any* is an NPI (3), while *already* is a PPI (4). NPIs and PPIs are said to be **polarity-sensitive**. Like degree, we treat polarity sensitivity as a continuous property on the [0,1] interval, where 0 is a very pronounced NPI, 1 a very pronounced PPI, with polarity-neutral items in the middle.

(3)     *Mary bought **any** books.          NPI
        Mary didn't buy **any** books.

(4)     John has arrived **already**.          PPI
        *John hasn't arrived **already**.

Sentences that are good contexts for NPIs and PPIs are said to have negative and positive polarity, respectively. Polarity of a sentence does not amount simply to the presence or absence of sentential negation, it is a way more complex semantic property (see Fauconnier 1975; Ladusaw 1979 and sub-

sequent literature). However, we will focus on the presence or absence of negation as a proxy to polarity in the current discussion.

Like degree, we will treat polarity sensitivity as a continuous property that fits into the [0, 1] interval, where 0 is a very pronounced NPI, 1 is a very pronounced PPI, with polarity-neutral expressions in the middle.

**Relation between the two properties**

Observations reported in linguistic literature suggest an interaction between these two properties (Israel, 1996, 2011; Solt, 2018; Solt and Wilson, 2021). The interaction is in many cases far from straightforward, but there are clear tendencies. Specifically, lower degrees associate with PPI behaviour. English degree modifiers support this observation (Solt and Wilson, 2021), as examples in (5) demonstrate. This pattern is found in other languages too (van Os, 1989; Nouwen, 2013; Ito, 2015). We will refer to modifiers of this lower degree range as Class v1 (low degree modifiers) later on.

(5)     The issue is **fairly / somewhat / rather / kind of / sort of** important.
        *The issue isn't **fairly / somewhat / rather / kind of / sort of** important.

Modifiers in the moderate range (Class v2, medium degree), to the contrary, show mild association with negative contexts (Israel, 1996). The association between negative polarity and degree modifiers from a certain range can most prominently be attributed to the phenomenon of 'negative strengthening' (Gotzner et al., 2018; Mazzarella and Gotzner, 2021):

(6)     John isn't **particularly** smart.

While the literal meaning of (6) is compatible with John being smart quite often these types of sentences are used to convey a stronger meaning: that John is not smart at all. This is a pragmatic asymmetry rather than a distributional constraint, but it contributes to the interaction patterns between degree and polarity sensitivity.

Finally, modifiers expressing very high degree (Class v3, high degree modifiers), behave like PPIs again:

(7)     This coffee is **damn / crazy** good.
        [??]This coffee isn't **damn / crazy** good.

---

[1]We use the vague and permissive term 'preference' here to cover the whole spectrum of asymmetries between positive and negative contexts that an expression shows – from ungrammaticality to decreased prominence of a narrow scope reading. Gradations of polarity sensitivity will play a crucial role in our discussion, but specifically for this reason we are looking for a unified way to describe the whole space of polarity sensitivity phenomena.

Existing work proposes analyses of degree modification with built-in causal connection between the degree semantics of modifiers and their polarity profile (Israel, 1996; Solt and Wilson, 2021) – even though the extent, exact shape and direction of this connection is not established yet. We use this state of affairs as a chance to contribute to this discussion empirically and analytically, using the method proposed below.

# 3 Method

In this section, we describe the details of a method to conduct artificial language learning experiments with pretrained LMs. Without loss of generality, we use BERT in our experiments, but other pretrained language models could be used instead.

We design our method to be applied to linguistic hypotheses of the form $A \Rightarrow B$, where $A, B$ are some properties in a given language. In this study, we specifically focus on the relationship between adverbial degree modification and polarity sensitivity. $A$ in this context is low, medium or high degree of an adverbial modifier $w$, and $B$ is negative, neutral or positive polarity of $w$. In general, we evaluate a hypothesis $A(w, i) \Rightarrow B(w, j)$ by showing that if $A$ holds according to BERT for token $w$ to an extent $i$, then so does $B$ to some extent $j$, according to BERT.

We use the cloze test (a task where the participant is asked to recover a missing language item) adapted for BERT (see Warstadt et al. 2019; Bylinina and Tikhonov 2022b for the cloze test on LMs for polarity). The test uses BERT's probability distributions over tokens in masked positions in diagnostic contexts for property $A$ or $B$.

To show that a hypothesis holds in general for an arbitatrary $w$, we:

(1) augment BERT's vocabulary with a set $W$ of new tokens and randomly initialize the corresponding embeddings;

(2) fine-tune the corresponding embeddings on a dataset where the new tokens appear in contexts that distributionally select for particular values of $A$;

(3) test whether the knowledge that $B$ holds was acquired, to the extent that follows the hypothesised association pattern with $A$.

As part of Step (1), we also verify that prior to training the initialized embeddings do not show any biases w.r.t. both properties $A$ and $B$. This approach presupposes a set of contexts that distributionally select for a specific linguistic property $X$, denoted $\mathcal{S}(X)$. We obtain such contexts by mining them from data, as described in Section 4.3. Part of future work is to extend the current method it to a more general case. The general structure of the synthetic dataset used to fine-tune the LM is described in Section 4.1. The dataset is also tailored to the linguistic phenomenon under investigation.

# 4 Dataset and measures

First, we delineate a fragment of English that will be the basis of our experiment (Section 4.1): simple sentences with a gradable adjective predicated over a definite noun phrase (as in *The pizza is good*). We re-shape these sentences to create diagnostic contexts for properties $A$ and $B$ (Sections 4.2, 4.3). We also use it to exemplify values of $A$ during training (Section 4.3).

## 4.1 Basic set of sentences

First, we automatically identified the set of gradable adjectives and nouns to build our training samples from. We started with `bert-base-uncased`[2] vocabulary and assigned all non-subword tokens a part of speech label with the SpaCy POS tagger[3]. We kept the top 1000 nouns. Using the CapitolWords dataset from `textacy`[4], we looked for co-occurrences of adjectives with degree modifiers *somewhat*, *very*, *really*, *extremely*, *rather* and picked 200 adjectives with the highest ratio of modified uses.

Second, we generated sentences with these nouns and adjectives using the following pattern:

$$\text{The } \texttt{noun}_x \text{ cop.PRS adj}_y$$

where `cop.PRS` is either singular or plural copula in the Present tense (*is* or *are*), $\texttt{noun}_x$ is one of the 1000 picked nouns, and $\texttt{adj}_y$ is one of the 200 gradable adjectives. The procedure gave us 400k sentences like these:

(8)   The purpose is interesting.
      The answer is simple.
      The environment is large.

---

[2] https://huggingface.co/bert-base-uncased

[3] https://github.com/explosion/spacy-models

[4] https://github.com/bdewilde/textacy-data

This 400k set varied in terms of naturalness, coherence and adherence to lexical selectional restrictions. To control for this, we ran the sentences through GPT-2[5] and kept the bottom 10k according to the assigned sentence perplexity.

The construction steps above aim to output 'natural' examples, based on insights from different sources (GPT-2, BERT, corpus-based statistics). Manual inspection of the resulting 10k dataset revealed some sentences that still sound intuitively 'weird'. We do not see this as a problem though, since the majority of sentences are natural enough.

The large quantity of examples in our dataset is crucial to make our experiments comparable to psycholinguistic experiments. In the latter, one sentence gives rise to a multiple of observations about (roughly) one linguistic system, due to judgements to multiple participants with similar enough intuitions. In our setting, we have only one agent (BERT), so we compensate by increasing the number of sentences.

## 4.2 Estimating polarity

To assign polarity scores to degree modifiers, we follow the procedure in (Warstadt et al., 2019; Bylinina and Tikhonov, 2022b). We use the 10k basic sentences (Section 4.1) to build a polarity contrast set. For each sentence in the basic set, a pair of sentences, one positive and one negative, with the [MASK] token in the modifier position:

> The $\text{noun}_x$ cop.PRS [MASK] $\text{adj}_y$.
> The $\text{noun}_x$ cop.PRS.NEG [MASK] $\text{adj}_y$.

We end up with 10k pairs of sentences like these:

(9)  The reason is [MASK] simple.
     The reason isn't [MASK] simple.

We use the generated sentence set to estimate polarity sensitivity *pol(m)* of a degree modifier *m* using the probabilities that BERT assigns to each token in its vocabulary in the masked position:

$$\frac{\sum_{s \in D}[p(\text{[MASK]} = m | s_{\text{POS}}^{mskd}) > p(\text{[MASK]} = m | s_{\text{NEG}}^{mskd})]}{|D|}$$

(1)

where $D$ is the 10k dataset, $s_{pos}^{masked}$ is a sentence $s$ from the dataset in the positive form, with *[MASK]* in the degree modifier position, and $s_{neg}^{masked}$ is its negative counterpart. So, we approximate polarity as the proportion of cases where token $m$ got a higher probability in *pos* than in *neg* context.

---

[5] https://huggingface.co/gpt2

Previous applications of this estimation method has shown its reliability for NPI detection (Jumelet and Hupkes, 2018; Warstadt et al., 2019; Jumelet et al., 2021; Bylinina and Tikhonov, 2022b). As an illustration, *slightly* gets a score of 0.99 (= is a PPI), *particularly* gets a score of 0.1 (is an NPI), while *incredibly* is a PPI again with score 0.94.

We use this polarity estimation method to get a reliable list of degree modifiers with polarity scores. For each of the 10k sentence pairs, we pick 100 tokens with highest probability in the masked position for a positive sentence and 100 tokens for its negative counterpart. Then we take two unions: one of all the "positive" tokens and one for the "negative" ones. We filter these two sets to only keep tokens that appear more than 100 times in one of them.[6] We use the resulting sets in the rest of the experiment.

## 4.3 Estimating and mining degree

To estimate polarity of tokens (Section 4.2), we relied on their patterns of occurrence in positive and negative contexts. To apply an analogous procedure to degree, we need contexts that associate with various degree semantics. We propose the following intuition. What does an answer to a yes/no-question with a gradable adjective – like *Is the pizza good?* – depend on? It certainly depends on how good the pizza is: the degree to which the property applies to the subject. Given that degree modifiers express exactly that, we can make a connection between their degree value and particles that answer the degree yes/no question.

For example, we expect particles to have different distribution in the masked position in (10) as an *effect* of the modifier:

(10)   – Is the pizza good?
       – [MASK], it is **somewhat** good.
       – [MASK], it is **extremely** good.

We use this idea to mine particles that are associated with low and high degree. The mined particles can be used to assess degree of the modifiers, analogously to polarity measurement above. As low degree modifiers, we use *somewhat* and *slightly*; for high degree, *very* and *extremely*. We modify each of the 10k sentences to generate pairs of sentences like these, where MOD is one of the four modifiers of interest:

---

[6] Among the tokens that survived the filter: *very, always, quite, so, really, too, all, actually*.

(11)   Is the question difficult?
       [MASK], it is MOD difficult.

As before, we run the resulting (40k) sentences through BERT and, for each sentence, we collect the top 100 tokens according to the probability of tokens in the masked position. We only keep those tokens that appear in this list 100 times or more. The particles in the resulting list are then tested their degree-diagnosing potential, as follows.

We use the same procedure as for polarity: for each particle, we check in what proportion of cases the probability that BERT assigns to the particle in the sentence with the high degree modifier is higher than with a low degree modifier. We perform this comparison for each of the four pairs of high vs. low degree modifiers: *very* vs. *somewhat*, *very* vs. *slightly*, *extremely* vs. *somewhat*, *extremely* vs. *slightly*. This procedure gives us a value from 0 to 1 for each particle from the list, depending on the extent to which it is associated with low degrees (the closer to 0, the more this holds) or high degrees (closer to 1). We fix the final set of top 10 particles that associate with **low** (12) degrees and with **high** degrees (13):

(12)   well, actually, now, but, however,
       still, so, why, anyway, sure

(13)   yes, oh, sir, absolutely, god,
       damn, remember, wow, seriously,
       man

Finally, we reverse the process and now use these particles to produce a degree score for degree modifiers. For each of the 10k sentences, we modify it to get 20 sentences like the following (where PRT ranges over the 20 particles in (12) and (13)):

(14)   Is the question difficult?   PRT,
       it is [MASK] difficult.

Comparing modifier probabilities across conditions defined by the distinction in (12) and (13) as before, we get a measure defined on the [0,1] interval that corresponds to the modifier's degree.

As a final step, we manually cleaned the resulting list of 415 tokens obtained from the [MASK] to get rid of syntactic junk and items whose selectional restrictions are too narrow, to end up with the list of 98 degree modifiers we will further use[7].

To validate our degree measure, we take five modifiers about which the literature agrees they introduce low or moderate degree (*barely*, *hardly*,

[7]Code and data are at https://github.com/altsoph/artificial_degree_modifiers

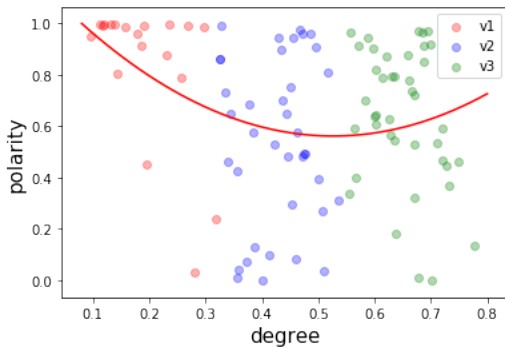

Figure 1: Degree and polarity of selected English modifiers. Estimated as described in Sections 4.2 and 4.3.

*rather*, *fairly*, *merely*); same for high degree (*completely*, *totally*, *utterly*, *damn*, *bloody*) (Paradis (1997); Bennett and Goodman (2018) a.o.). There's no overlap with modifiers we used as seeds for our degree measure. Also, these are practically **all** modifiers with an undisputed degree profile discussed in the literature that are also whole BERT tokens. Our measure assigns the low degree class an average score of 0.22 (min 0.11; max 0.31); average of 0.56 for the high degree class (min 0.5; max 0.62). *Very*, which has some intensifying effect got a score of 0.45. We conclude that the measure is decent, albeit somewhat shifted to the left.

Fig. 1 shows the distribution of polarity sensitivity and degree for these modifiers (we color-code them as moderate, medium and high degree). As the scatterplot and the fitted parabola show, the existing data is compatible with what is hypothesised in the linguistic literature: low degrees associate with positive polarity, while the rest is more varied – mid-range degrees gravitate towards more negative polarity somewhat, while the higher range again gravitates towards PPI behaviour.

### 4.4  Degree and polarity in BERT embeddings

We use diagnostic classifiers to analyse how polarity sensitivity and degree semantics are represented in BERT token embeddings for degree modifiers. Using embeddings of degree modifiers as features, we fit logistic regression with L1 regularization to demote non-zero coefficients for two binary classification tasks: 1) token classification into 'negative' ($< .5$) and 'positive' ($> .5$) with respect to polarity; 2) token classification into 'low degree' ($< .4$, based on somewhat skewed score distribution) and 'high degree' ($> .4$).

On 5 folds, average accuracy for polarity on train data is 79.2%, and 74.7% on test. For degree, it's 73% and 72.3%, respectively. For each of the tasks,

we find the most important part of the embedding that is responsible for the distinction, by taking coordinates that have non-zero coefficients in at least four of the folds. We found 20 important coordinates for polarity and 13 for degree. There was no overlap between these coordinates, indicating no representational overlap between polarity and degree at the level of token embeddings. If it turns out that the model encodes the dependency between the two properties, it would be on a level other than embeddings directly.

## 5 Experiment

This section describes how we teach BERT a new system of degree modifiers. Section 5.1 describes how we introduced new tokens into BERT's vocabulary by using particles that signal the properties we wish to teach BERT. Section 5.2 provides the details of the fine-tuning procedure and the experimental results.

### 5.1 Mining contexts for new degree modifiers

We partition the existing degree modifiers into three same-sized groups, based on the degree scale region they belong to (according to degree estimation procedure in Section 4.3): moderate, medium, high (or, v1, v2 and v3, respectively). The groups are three color-coded vertical divisions in Fig. 1. We use the identified groups to instantiate three classes of new degree modifiers. For each of the groups, we mine degree-region-specific particles, using the procedure in Section 4.3. The resulting sets of 3-way degree-diagnosing particles are:

```
v1: alternatively, myself, similarly,
    accordingly, otherwise, however,
    alternately, likewise, conversely,
    er, although, thus, nevertheless,
    nonetheless, still, hence

v2: yes, once, naturally, evidently,
    eventually, not, surely, nowadays,
    however, someday, fortunately, here,
    presumably, ideally, accordingly,
    hopefully

v3: god, gods, goddess, dammit, christ,
    goddamn, jesus, fucking, holy, kate,
    damn, skyla, lord, princess, love,
    daddy
```

For each of the three groups, we instantiate 33 new modifiers. Then, for each sentence in the 10K set, we generate a v1 sentence, a v2 and a v3. The sentences are of the same question-answer form as in Section 4, and in each of them we insert a randomly

| | Before training | | After training | |
|---|---|---|---|---|
| | degree | polarity | degree | polarity |
| v1 | 0.48, 0.06 | 0.42, 0.24 | 0.18, 0.02 | 0.99, 0.03 |
| v2 | 0.50, 0.06 | 0.43, 0.21 | 0.40, 0.02 | 0.00, 0.00 |
| v3 | 0.48, 0.06 | 0.39, 0.18 | 0.83, 0.02 | 0.85, 0.26 |
| **Baselines** | | | | |
| random | 0.52, 0.06 | 0.38, 0.20 | 0.41, 0.09 | 0.83, 0.30 |
| untrained | 0.50, 0.06 | 0.39, 0.20 | 0.42, 0.08 | 0.00, 0.00 |

Table 1: Estimates of polarity and degree of new tokens before and after training. Each pair of numbers represents a mean and a standard deviation. v1, v2, v3 represent polarity and degree statistics for the new modifiers (low, medium, high) from our main experiment.

picked particle corresponding to the degree class of the modifier ($n$ = number id):

(15)
```
Is the reason simple?  [prt_v1],
it is [mod_v1_n] simple.
Is the reason simple?  [prt_v2],
it is [mod_v2_n] simple.
Is the reason simple?  [prt_v3],
it is [mod_v3_n] simple.
```

### 5.2 Fine-tuning BERT to new tokens

We split the dataset into training and validation parts with 0.85:0.15 ratio. Then we randomly mask 15% of tokens in the resulting dataset and fine-tune BERT for the task of masked token prediction. We use the same type of pretrained BERT model as in the previous steps. We use the Adam optimization algorithm with decoupled weight decay regularization (Kingma and Ba, 2014; Loshchilov and Hutter, 2017) and learning rate of `5e-5`. We use the batch size of 32 and fine-tune the model for three epochs. For the training, we freeze all weights except for the very first layer of token embeddings.[8]

We compare our method against two baselines:

- **random baseline:** 99 randomly initialized tokens are trained in contexts with particles randomly chosen from any of the three sets (v1, v2 and v3);

- **untrained baseline:** 99 new tokens to be randomly initialized before the training phase, but not fine-tuned.

Upon fine-tuning, the three groups of tokens form three clusters, as shown in Fig. 2. Tokens that

---

[8]This decision is based on the intuition that learning new words in an artificial language learning setting shouldn't lead to deep changes in prior linguistic knowledge of a native language for a realistic learner.

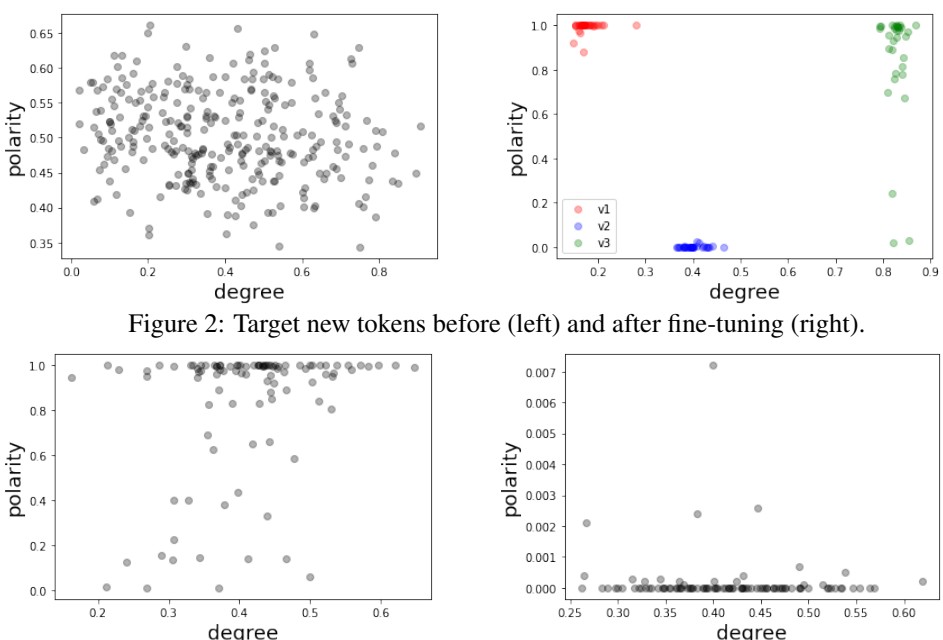

Figure 2: Target new tokens before (left) and after fine-tuning (right).

Figure 3: Baselines: contexts randomly mixed during training (left) and untrained tokens (right)

belong to groups v1 and v3 cluster in the PPI region, medium-degree tokens (v2) show NPI-like behaviour. This is generally in line with linguistic observations described in Sections 2 and 4. The two baselines (Figure 3), as expected, do not show pronounced degree profiles – but develop non-random polarity behaviour. The random baseline gravitates towards positive polarity, while the untrained baseline shows NPI behaviour. Means and standard deviations for degree and polarity before and after training are listed in Table 1.

## 6 Discussion and future work

### 6.1 Interpretation of the experimental results

We saw that the training organized the new tokens into three clusters. First, we observe that the tokens develop low, medium or high degree behaviour, as intended by dataset construction. This means that our procedure conveyed degree information to the model. Furthermore, polarity scores upon training show that the three groups generally follow the hypothesis from Section 2 and analysis from Section 4.3: low and high degrees lead to PPI behaviour, while medium degrees are associated with negative polarity.

What is somewhat surprising though is how strong the association with negative polarity is for medium degrees. Here, looking at our baselines might provide a hint towards an explanation. The random baseline develops PPI behaviour: this is not particularly surprising given that a random pool

of degree contexts is bound to contain a majority of PPI-associated diagnostic particles (this holds for both low and high degree, that is, 2/3 of datapoints). So, the model has prevailing evidence to treat random baseline items as PPIs. Untrained baseline is more interesting in this respect: new tokens that did not appear in the training dataset at all develop NPI behaviour. We do not know what leads to this, but, at the level of observation, a general shift in the direction of lower polarity scores for the whole lexicon might be some artefact of our training procedure. If this is true, the very low polarity scores that we see for some items should be interpreted as actually corresponding to somewhat higher scores. We leave exploration of this effect to future work.

### 6.2 Limitations and future work

Summing up Sections 5.2 and 6.1, our results are compatible with existing linguistic observations concerning the relation between degree and polarity. However, the biggest question to our approach is how much we can trust the obtained results in making conclusions about natural language. We could gain insight on this question by reproducing the experiment with human participants. The experiment with artificial LMs could serve as a preliminary step to polish the underlying hypothesis and the setup for the human experiment. We leave to future work as well.

Another question is whether there is a reliable way to introduce property $A$ without leaking information about property $B$ in the training data.

Admittedly, the simple procedure we follow does not take specific precautions to convincingly show this did not happen. We hope that the version of the experiment that we present here will serve as a starting point for future work developing methods to address this question or recycling existing tools from other types of experiments.

## 7 Conclusion

We introduced a method to assess the connection between two linguistic properties as encoded by pre-trained neural LMs. We applied this method to an observation in linguistic semantics: the relation between degree and polarity sensitivity. We found that the experimental results are in line with the generalizations from the linguistic literature, indicating validity of our approach and pointing in the direction of BERT making the generalization in question. We hope that this set-up can be applied to other types of models (trained on languages other than English, or multilingual) and other linguistic generalizations, both within individual languages and cross-linguistically.

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
