# OpenReview forum: "Connecting degree and polarity: An artificial language learning study"
_EMNLP/2023/Conference — EMNLP 2023 Main_

### Official Review · Reviewer_DQ6w · 2023-07-28

**Soundness:** 4

**Excitement:**

4: Strong: This paper deepens the understanding of some phenomenon or lowers the barriers to an existing research direction.

**Missing References:**

- Denić, Milica, Vincent Homer, Daniel Rothschild, and Emmanuel Chemla. (2021). "The influence of polarity items on inferential judgments." Cognition
- Diagnostic Classifiers (Section 4.4) are not cited, cite e.g. Hupkes et al. or Belinkov here.

**Paper Topic And Main Contributions:**

The paper investigates the connection between two linguistic properties: polarity-sensitivity and degree modification. Polarity sensitive items are words like *any*, which can only occur within a negative context (I didn't/*did buy any books). Degree modifiers are words like *slightly* and *very*. The connection between these two properties is that low-degree modifiers tend to support positive polarity items (*The issue is fairly important*), whereas high-degree modifiers tend to support negative polarity items (*John isn't particularly smart*).

This connection is investigated in the context of LMs, in particular BERT. BERT is employed as a learning agent here: via an elaborate experimental setup, the authors attempt to see how the connection between polarity<>degree manifests itself within BERT, and how BERT generalises to novel tokens for which it has no prior knowledge about degree or polarity.

**Questions For The Authors:**

1. Around L359 it is mentioned that BERT acts as only one "psycholinguistic subject". Could this be addressed by running your experiments on a larger suite of LMs, for example the 25 MultiBERT models?
2. Are the standard deviations in Table 1 over multiple seeds of finetuning?
3. I genuinely do not understand how your finetuning setup would lead to the v2 group to result in a polarity score of 0.0 (Table 1). What is different about these tokens to result in such a drastically different outcome as the v1/v3 groups? I see you address this a bit in Section 6, but I don't think it's fair to leave this interpretation open for future work.
4. You state in section 2 the the connection between degree and polarity is somewhat linear (low-degree+PPI, high-degree+NPI), but in Figure 1 and around L622 you state that your parabolic finding is in line with prior linguistic research. Do I interpret this wrong, or is there a contradiction here?

**Reasons To Accept:**

- Strong background on linguistic principles that makes it well-suited for a venue like EMNLP.
- Exciting usage of LMs as linguistic learning agents, which allows for testing of complex linguistic hypotheses in a controlled environment. The setup described in this paper will definitely inspire other linguistics-minded NLP researchers.

**Reasons To Reject:**

- The paper could be structured a bit more clearly, which makes it hard to follow at times. E.g., the setup described in Section 3 only plays a role three pages later, and it took me a while to figure out how everything relates together.
- The main results present a somewhat confusing pattern, which is not fully addressed by the analysis either.

**Reproducibility:**

4: Could mostly reproduce the results, but there may be some variation because of sample variance or minor variations in their interpretation of the protocol or method.

**Reviewer Confidence:**

4: Quite sure. I tried to check the important points carefully. It's unlikely, though conceivable, that I missed something that should affect my ratings.

**Typos Grammar Style And Presentation Improvements:**

## Suggestions
- I would suggest using DeBERTa instead of BERT, which is generally speaking a better model (but I can see that the exact model choice is not vital at all for the merit of the paper)
- In prior work it has been shown that using randomly initialised embeddings may be suboptimal for pretrained LMs, and that it is better sample from the manifold of the existing embeddings instead. See for example Papadimitriou & Jurafsky (2023), section 5.1.2: https://arxiv.org/pdf/2304.13060.pdf
- It would be nice to report the results of the experiment in 4.2 somewhere (small table or in appendix). Also: is it possible to evaluate the BERT scores with some human data from elsewhere? I don't know if such a resource exist, but I find this experiment interesting in itself already.
- If I understand section 4 correctly, the experiments therein show rather the extent to which the polarity<>degree hypothesis holds within vanilla BERT. I would emphasize this a bit more, the introduction made me expect that all experiments would focus on artificial language learning.
- The parabolic relationship between degree<>polarity in Fig. 1 came as a surprise to me, I suggest explaining the expected behaviour a bit more clearly in Section 2 already (see also my Q4).
- At L491 it is stated that the max degree score is 0.62, but in Fig. 1 the degree axis ranges to 0.8. Is the information at L491 correct, or should I not interpret these numbers as being the same?
- Section 4.4 does not add a lot to the story in my opinion, and could be shortened.

## Style
- Represent the mean,std tuples in Table 1 as mean ± std instead. (I personally prefer using a slightly smaller font for the std values also to make the mean values stand out a bit more, since those are most important here).

## Typos
- l27: relation -> relations
- l440: tested *for* their degree-diagnosing
- l636: *The* untrained baseline
- l658: We leave *this open* to future work
- l678: indicating *the* validity

---

> ### Author Rebuttal · Authors · 2023-08-25
>
> We would like to thank the reviewer for the deep, friendly and fair discussion. This is exactly the kind of discussion we were hoping for when working on this!
>
> Answers to the Reviewer's questions:
>
> 1. Around L359 it is mentioned that BERT acts as only one "psycholinguistic subject". Could this be addressed by running your experiments on a larger suite of LMs, for example the 25 MultiBERT models?
>
> That's a great idea, actually. That'd be a pretty computationally heavy experiment, but we think it's worth it.
>
> 2. Are the standard deviations in Table 1 over multiple seeds of finetuning?
>
> Yes.
>
> 3. I genuinely do not understand how your finetuning setup would lead to the v2 group to result in a polarity score of 0.0 (Table 1). What is different about these tokens to result in such a drastically different outcome as the v1/v3 groups? I see you address this a bit in Section 6, but I don't think it's fair to leave this interpretation open for future work.
>
> The shortest answer is: We don't know. A slightly longer answer is: That's kind of the point -- we expected the parabolic outcome with respect to polarity, and that's what we find. Why such an exaggerated pattern? Can it be that the token embeddings for this group of modifiers under-trained and thus closer to untrained baseline? Here initializing the embeddings non-randomly would give a hint, per reviewer's helpful suggestion (when we conducted the experiments, the 2023 paper wasn't yet out)! Maybe we have time to test it for the longer version of this paper, if accepted. (In fact, we found the comments in this review so helpful and on-point that we would've been happy to continue further investigations in collaboration with the reviewer, but the anonymity..)
>
> 4. You state in section 2 the the connection between degree and polarity is somewhat linear (low-degree+PPI, high-degree+NPI), but in Figure 1 and around L622 you state that your parabolic finding is in line with prior linguistic research. Do I interpret this wrong, or is there a contradiction here?
>
> It's true that clarity of Section 2 fell victim of dramatic shortening of the text before submission! Basically, it does not state the direction or linearity of the relation between degree and polarity, now it just says that there are some interactions that linguists noticed. In fact, it should say it is parabolic, as linguists know. We could also have done a better job mapping the degrees of the modifiers we discuss in Section 2 to the degrees we work with later in the experimental part (v1, v2, v3). Low-to-moderate is v1, moderate-to-high is v2. Overall, the reviews showed us weak points in how the paper is structured and some things are explained. We will streamline the paper and do a better job at presentation, if accepted.
>
> Re other suggestions:
>
> 1. Missing references. We know this work (the first paper was actually cited in the penultimate draft which was shortened and the paper didn't make it to the final version..) and will cite it in the next version of the paper for sure, especially when given the extra page.
>
> 2. Parallel study with human participants: a great idea, something that we also want to do -- waiting for the right funding moment.
>
> 3. "At L491 it is stated that the max degree score is 0.62, but in Fig. 1 the degree axis ranges to 0.8." At L491, the max degree score refers to the hand-picked sanity-check subset of modifiers, not the whole set. Will make this more clear in the text.

---

### Official Review · Reviewer_tJwV · 2023-08-01

**Soundness:** 3

**Excitement:**

3: Ambivalent: It has merits (e.g., it reports state-of-the-art results, the idea is nice), but there are key weaknesses (e.g., it describes incremental work), and it can significantly benefit from another round of revision. However, I won't object to accepting it if my co-reviewers champion it.

**Paper Topic And Main Contributions:**

The paper investigates the sensitivity of a pre-trained language model on degree modifiers and sentence polarity. In particular, the authors hypothesize that the degree expressed by a modifier (low, medium, or high degree) is related to the modifier's sensitivity to sentence polarity. To test this hypothesis, they apply the Artificial Language Learning experimental paradigm from psycholinguistics to the BERT model. The experimental results suggest that BERT generalizes in accordance with linguistic observations, associating degree semantics with polarity sensitivity, particularly low degree semantics being associated with positive polarity sensitivity.

**Questions For The Authors:**

How was the process of adding new tokens to the model vocabulary carried out?

**Reasons To Accept:**

-- The application of the "Artificial Language Learning" paradigm from psycholinguistics to study linguistic generalizations in pre-trained language models is a compelling approach. This methodology adds value to the paper by offering new insights into how language models handle degree modifiers and their relation to sentence polarity.

-- The authors provide a comprehensive and detailed description of the studies conducted on degree modifiers and polarity sensitivity in Section 2. This helps readers understand the context and motivation for their experiments.

**Reasons To Reject:**

-- Some parts of the paper are complicated, and several steps in the methodology are not adequately explained. For instance, the process of introducing new tokens into BERT's vocabulary needs clearer discussion. The authors should provide a more explicit description of how these new tokens were added to the model's vocabulary, especially concerning how the weights associated with the selected degree modifiers were initialized.

-- The results discussion section appears to be brief and lacking in-depth analysis. The paper would benefit from a more concise description of the methodology, allowing more space for analyzing and discussing the obtained results. A clearer presentation of the results would enrich the paper and stimulate further thought and interpretation. I would also suggest to restructure the paper as a short paper.

-- As I wrote previously, Section 2 is interesting for what concerns the description of studies related to degree and polarity sensitivity, but it lacks context from more NLP-oriented studies. Including related work on similar approaches applied to study language model competencies would strengthen the paper's background and demonstrate its relevance within the broader NLP research.

-- While it is not necessary to test a phenomenon on an extensive range of models, studying interpretability in language models could benefit from testing the approach on a few additional models. Incorporating at least one or two other models and evaluating performance on different languages would bolster the claims made in the paper.

**Reproducibility:**

2: Would be hard pressed to reproduce the results. The contribution depends on data that are simply not available outside the author's institution or consortium; not enough details are provided.

**Reviewer Confidence:**

4: Quite sure. I tried to check the important points carefully. It's unlikely, though conceivable, that I missed something that should affect my ratings.

---

> ### Author Rebuttal · Authors · 2023-08-25
>
> We thank the reviewer for the detailed analysis of the paper and for pointing out its weaker aspects that might need attention from our side.
>
> Answer to the question the reviewer poses:
>
> Question: How was the process of adding new tokens to the model vocabulary carried out?
> Answer:
> We use standard techniques to introduce new tokens, as provided by Huggingface API: https://huggingface.co/docs/transformers/internal/tokenization_utils#transformers.SpecialTokensMixin.add_tokens
> Technically, the process consists of several steps:
>  * add the new token to token2id map in the tokenizer
>  * resize the embedding tensor in the model by adding a corresponding number of columns
>  * initialize them by random weights
> There are more complex strategies (see for example, arxiv:2112.14569), but we used the currently-standard practice.
> The random initialization of these weights is stated in the paper twice (L100-101, L287-289), the code to replicate the our whole experimental pipeline, including the new tokens addition, can be found in the anonymized github repo that we provide a link to in the paper.
>
>
>
> Comments on Reasons To Reject:
>
> -- Some parts of the paper are complicated, and several steps in the methodology are not adequately explained. For instance, the process of introducing new tokens into BERT's vocabulary needs clearer discussion. The authors should provide a more explicit description of how these new tokens were added to the model's vocabulary, especially concerning how the weights associated with the selected degree modifiers were initialized.
>
> We agree that some parts of the paper would benefit from streamlining and better presentation, thanks to the reviewers' comments! We will make sure to describe what we did in a clearer way. But also want to re-iterate that the random initialization of weights is stated in the paper twice, and the code for new token initialization is available on GitHub, with the repository linked in the paper. We would like to note that a Reproducibility score 2 in the situation when there is a GitHub repository with the whole set of ingredients to reproduce the results, seems like a bit of a stretch even if some descriptions of the method in the text could use a bit of restructuring.
>
> -- The results discussion section appears to be brief and lacking in-depth analysis. The paper would benefit from a more concise description of the methodology, allowing more space for analyzing and discussing the obtained results. A clearer presentation of the results would enrich the paper and stimulate further thought and interpretation. I would also suggest to restructure the paper as a short paper.
>
> It would be great to have more space to discuss the results! Doing it by shortening the methodology would contradict the previous suggestion, however, requesting higher level of detail in the methodology part. In any case, none of this would be possible if the paper is restructured as a short paper, so the requests seem internally contradictory.
>
>
> -- As I wrote previously, Section 2 is interesting for what concerns the description of studies related to degree and polarity sensitivity, but it lacks context from more NLP-oriented studies. Including related work on similar approaches applied to study language model competencies would strengthen the paper's background and demonstrate its relevance within the broader NLP research.
>
> We would love to learn about other NLP-oriented work with similar focus and experimental methods, and we will happily include citations/discussions of this work in future versions of the paper.
>
> -- While it is not necessary to test a phenomenon on an extensive range of models, studying interpretability in language models could benefit from testing the approach on a few additional models. Incorporating at least one or two other models and evaluating performance on different languages would bolster the claims made in the paper.
>
> We agree that adding more models and more languages can enrich the conclusions we draw. We disagree that this is a reason for rejection.

---

### Official Review · Reviewer_zJs2 · 2023-08-11

**Typos Grammar Style And Presentation Improvements:** 1. In the Equation (1), there is s^{m…
**Soundness:** 4

**Excitement:**

4: Strong: This paper deepens the understanding of some phenomenon or lowers the barriers to an existing research direction.

**Paper Topic And Main Contributions:**

The authors presented a technique to evaluate the correlation between the degree of modifiers and their polarity distribution as represented by pre-trained LLMs. The outcomes of our experiments aligned with established linguistic findings, confirming the soundness of their approach and suggesting the potential for BERT to make generalizations. There are several potential contributions: (1) they provided another case of linguistic evaluation, confirming LLMs' good linguistic knowledge even when the knowledge is quite subtle and not staying at the morphological syntactic level; (2) they provided novel techniques for evaluation -- in addition to common practices in psycholinguistic research, their mining procedure could be used in further studies and is quite inspiring; (3) the final fine-tuning and generalization procedure could also attract more attention in the field to the question of learning in general.

**Questions For The Authors:**

1. If you expand the proxies for polarity to cues that are not just the presence of negation, what knowledge would you derive further?

### Follow-up: It would be more helpful if you could find relevant citations on the other types of contexts in "Looking at other types of contexts, ... contexts with indirect polarity marking.".

2. The example (6) and its explanation aren't particularly clear to me. Are you suggesting that modifiers with moderate-to-high range don't pattern with negation? If so, together with (5), are you saying only mid-range modifiers pattern with negation? Could you provide another example?

### This part is solved.

**Reasons To Accept:**

Besides the above-mentioned contributions, the clear presentation of the paper could lead the readers to a good learning experience: the presentation of the linguistic facts is clear, concise and to the point; the mining and experiment procedure was stated clearly with great potential for replication and extension.

The authors have good knowledge of existing research in linguistics and NLP/NLU. They are able to synthesize the insights and delineate a clear picture of their contribution.

The overall high-level research question is appealing and worthwhile -- we are not only interested in the evaluation of the models' current abilities but also in the generalization ability as well as the learning question. I think the methodology the authors used in this paper could be applied to more case studies to bring us closer to the major learning question.

**Reasons To Reject:**

I don't have particular reasons to reject this paper, but I would appreciate learning more about why the authors picked this specific phenomenon against the high-level research question.

**Reproducibility:**

3: Could reproduce the results with some difficulty. The settings of parameters are underspecified or subjectively determined; the training/evaluation data are not widely available.

**Reviewer Confidence:**

5: Positive that my evaluation is correct. I read the paper very carefully and I am very familiar with related work.

---

> ### Author Rebuttal · Authors · 2023-08-25
>
> We are very grateful to the reviewer for the positive comments! There is not much material for rebuttal here, since we seem to be on the same page about the aspects of the work that make it strong, as well as the weaker sides of how we present it (the paper does need streamlining, and thanks to the reviewers, we now see the exact points where this is needed).
>
> Answers to the questions:
>
> 1. If you expand the proxies for polarity to cues that are not just the presence of negation, what knowledge would you derive further?
>
> That's a great question. Polarity (negative vs. positive) can be analysed in different ways. A superficial notion of polarity could just be the presence or the absence of overt negation. Or it could be a deeper grammatical notion that encompasses the presence vs. absence of negation, but also, additionally, shows up in other contexts marked by other, more subtle, polarity clues (quantifiers 'no' vs. 'some', 'few' vs. 'many' and so on -- a very long list, in fact, arguably, infinite). Looking at negation only, we cannot disentangle which notion of polarity is at play in the cases we study. Looking at other types of contexts, we could see whether there is a qualitative difference between the results we report here and results on contexts with indirect polarity marking. That would give us a clearer idea on the underlying generalisations that the models encode.
>
> 2. The example (6) and its explanation aren't particularly clear to me. Are you suggesting that modifiers with moderate-to-high range don't pattern with negation? If so, together with (5), are you saying only mid-range modifiers pattern with negation? Could you provide another example?
>
> Section 2 is definitely not a presentational success on our part, thanks for pointing it out! The distilled message of this section is simply that polarity and degree interact in a non-trivial way. It should say instead that the linguists know that the relation is kind of parabolic, with the lower and high end on the degree scale being positive polarity items, the middle actually having a tendency of combining with (and being interpreted in the scope of) negation to the extent other degree groups don't. There's much more to say about it and the reasons behind it (for instance, extreme modifiers, expressing very high degrees, are 'expressives' in the technical sense of the word: always getting the highest possible scope and thus avoiding being embedded under anything, including negation -- see, for instance, a meta-linguistic feel that a sentence like 'That van is not fricking huge' gives) -- but we can't spend too much space on this. What we should definitely do is a better job mapping the degrees of the modifiers we discuss in Section 2 to the degrees we work with later in the experimental part (v1, v2, v3). Low-to-moderate is v1, moderate-to-high is v2. We should also add a very high degree modifier as an example.
>
> Overall, with the help of the reviewers we identified the weak points in how we present our set-up and results (Section 5 is another example of what should be improved) and we will introduce the respective changes in the new versions of the paper.

---

### Meta-Review · Area_Chair_gSVQ · 2023-09-18

**Recommendation:** 4

**Metareview:**

The paper introduces a method to evaluate the relationship between degree expressed by modifiers and their sensitivity to sentence polarity. The authors draw on the artificial language learning paradigm from psycholinguistics and apply it to BERT, finding that the model’s generalization patterns align with linguistic observations relating degree and polarity sensitivity.

The reviewers express that the paper is clear and presents a meaningful synthesis between linguistics and NLP. The reviewers also express that the methodological setup is novel, interesting, inspiring. The reviewers raised some issues pertaining to clarity, much of which has been addressed in discussion.

My main hesitation, also mentioned in some of the reviews, is the focus of the paper’s experiments solely on the BERT model, which imposes some limitations on the generality and impact of the conclusions. However, based on the merits and interest of the methodology, and the interdisciplinary value, I recommend that the paper be accepted at least to Findings, and potentially to the main conference.

---

### Decision · Program_Chairs · 2023-10-07

**Decision:**

Accept-Main

**Comment:**

The paper introduces a method to evaluate the relationship between degree expressed by modifiers and their sensitivity to sentence polarity. The authors draw on the artificial language learning paradigm from psycholinguistics and apply it to BERT, finding that the model’s generalization patterns align with linguistic observations relating degree and polarity sensitivity.

The reviewers express that the paper is clear and presents a meaningful synthesis between linguistics and NLP. The reviewers also express that the methodological setup is novel, interesting, inspiring. The reviewers raised some issues pertaining to clarity, much of which has been addressed in discussion.

My main hesitation, also mentioned in some of the reviews, is the focus of the paper’s experiments solely on the BERT model, which imposes some limitations on the generality and impact of the conclusions. However, based on the merits and interest of the methodology, and the interdisciplinary value, I recommend that the paper be accepted at least to Findings, and potentially to the main conference.